# Design and Characterization of Myristoylated and Non-Myristoylated Peptides Effective against *Candida* spp. Clinical Isolates

**DOI:** 10.3390/ijms23042164

**Published:** 2022-02-16

**Authors:** Francesca Bugli, Federica Massaro, Francesco Buonocore, Paolo Roberto Saraceni, Stefano Borocci, Francesca Ceccacci, Cecilia Bombelli, Maura Di Vito, Rosalba Marchitiello, Melinda Mariotti, Riccardo Torelli, Maurizio Sanguinetti, Fernando Porcelli

**Affiliations:** 1Dipartimento di Scienze Biotecnologiche di Base, Cliniche Intensivologiche e Perioperatorie, Università Cattolica del Sacro Cuore, 00168 Rome, Italy; francesca.bugli@unicatt.it (F.B.); wdivit@gmail.com (M.D.V.); rosalba.marchitiello01@unicatt.it (R.M.); melinda.mariotti@unicatt.it (M.M.); 2Dipartimento di Scienze di Laboratorio e Infettivologiche, Fondazione Policlinico Universitario A, Gemelli IRCCS, 00168 Rome, Italy; riccardo.torelli@policlinicogemelli.it; 3Department for Innovation in Biological, Agrofood and Forest Systems, University of Tuscia, 01100 Viterbo, Italy; federica.massaro@studenti.unitus.it (F.M.); fbuono@unitus.it (F.B.); paoloroberto33@gmail.com (P.R.S.); borocci@unitus.it (S.B.); 4CNR—Institute for Biological Systems, Area Della Ricerca di Roma 1, SP35d 9, 00010 Montelibretti, Italy; 5CNR—Institute For Biological Systems, Sede Secondaria di Roma-Meccanismi di Reazione, c/o Università La Sapienza, 00185 Rome, Italy; francesca.ceccacci@cnr.it (F.C.); cecilia.bombelli@cnr.it (C.B.)

**Keywords:** lipopeptide, antifungal activity, myristoylation, in vivo infection control

## Abstract

The increasing resistance of fungi to antibiotics is a severe challenge in public health, and newly effective drugs are required. Promising potential medications are lipopeptides, linear antimicrobial peptides (AMPs) conjugated to a lipid tail, usually at the N-terminus. In this paper, we investigated the in vitro and in vivo antifungal activity of three short myristoylated and non-myristoylated peptides derived from a mutant of the AMP *Chionodracine*. We determined their interaction with anionic and zwitterionic membrane-mimicking vesicles and their structure during this interaction. We then investigated their cytotoxic and hemolytic activity against mammalian cells. Lipidated peptides showed a broad spectrum of activity against a relevant panel of pathogen fungi belonging to *Candida* spp., including the multidrug-resistant *C. auris*. The antifungal activity was also observed vs. biofilms of *C. albicans*, *C. tropicalis*, and *C. auris*. Finally, a pilot efficacy study was conducted on the in vivo model consisting of *Galleria mellonella* larvae. Treatment with the most-promising myristoylated peptide was effective in counteracting the infection from *C. auris* and *C. albicans* and the death of the larvae. Therefore, this myristoylated peptide is a potential candidate to develop antifungal agents against human fungal pathogens.

## 1. Introduction

In recent years, extensive abuse of chemicals to control diseases has been associated with the increasing number of multidrug-resistant pathogens. The growth of fungi and bacteria resistant to conventional antimicrobial agents represents one of the biggest concerns worldwide [1,2,3,4].

Systemic fungal infections are challenging and represent a significant cause of morbidity and mortality, especially for immune-compromised people. Fungi infect billions of people every year, with a high rate of mortality of about 1.5 million [1]. Pathogenic fungi, like human cells, are eukaryotic, and potential selective drugs can only attack limited targets. Many antifungal drugs act on the biosynthetic pathway of ergosterol, the functional analog of cholesterol in animal cells, essential for regulating membrane fluidity, or on the cell wall component 1,3-β-D-glucan, or internal components, such as fungal nucleic acids [2,3]. However, the only licensed antifungal therapies for humans, polyenes, azoles, echinocandins, and flucytosine, suffer several limitations due to cross-target toxicity. Thus, the need for more effective and less toxic antifungal agents is irrefutable. Antimicrobial peptides (AMPs), which are key components of the innate immune system [4], are ubiquitously present in vertebrates, invertebrates, plants, and bacteria and represent promising antifungal candidates. In 1948, Bacillomycin, the first AMP with antifungal activity, was isolated from the bacteria *Bacillus subtilis* [5] and, in recent years, others were evidenced [6,7].

An important class of AMP is constituted in its mature form by 15–50 amino acid residues; they are generally cationic and adopt an amphipathic structure upon interacting with the microbial negatively charged membrane. α-helices and β-sheets are the most common secondary structures they adopt [8,9,10]. AMPs typically kill pathogens by disrupting the cell membrane with consequent leakage of internal material [11], but they also could enter the pathogen cell and reach an intracellular target [12,13]. Moreover, AMPs can represent a valid alternative to overcome the antimicrobial resistance to conventional antibiotics because they are less susceptible to the evolution of resistance in microorganisms [14,15]. However, most studies on AMPs are focused on their antibacterial properties and not on their antifungal activity. Interestingly, AMPs could also prevent fungal biofilm formation or eradicate preformed biofilms; their efficacy has been associated with different mechanisms of action, such as cell wall perturbation, the inhibition of cell adhesion onto surfaces, and interaction with other targets inside fungal cells [16].

An interesting approach to increase the antifungal activity of AMPs is to conjugate small peptides with fatty acids of different lengths. Natural lipopeptides, produced in bacteria and fungi, are usually composed of a fatty acid chain conjugated to a short linear or cyclic peptide. In 2003, the Food and Drug Administration (FDA) approved Cubicin^R^ (Daptomycin) as the first cyclic lipopeptide antibiotic for the treatment of severe infections caused by Gram-positive bacteria, despite its toxicity [17]. The proposed mechanism of action involves cyclic chain insertion into the cell membrane with consequent depolarization and ionic efflux. Various studies demonstrated that the conjugation of a cationic linear antimicrobial peptide with a fatty acid chain could improve its antimicrobial and antifungal activity [18,19,20,21]. Indeed, lipopeptides with positively charged residues retain the amphipathic structure typical of AMPs, and the presence of the hydrophobic tail enhances their surface-active properties.

In this paper, starting from the template of *KHS-Cnd*, a 22-amino-acid residue mutant of the AMP *Chionodracine* [22], we designed a series of 11-residue peptides. The three peptides WFGKLYRGITK (Pep-A), WRGITKVVKKV (Pep-B), and WVVKKVKGLLK (Pep-C) have been conjugated at the N-terminus with myristic acid to obtain three lipopeptides, Myr-WFGKLYRGITK (Myr-A), Myr-WRGITKVVKKV (Myr-B), and Myr-WVVKKVKGLLK (Myr-C). All the peptides have been characterized by CD spectroscopy, and their self-assembly properties have been elucidated. Molecular dynamics (MD) simulations were carried out to investigate, with an atomic resolution, the interaction of myristoylated and non-myristoylated peptides with a zwitterionic membrane model (POPC) and an anionic membrane model (POPC/POPG). Furthermore, studies carried out with membrane-mimicking systems revealed the ability of lipopeptides to interact with both zwitterionic and anionic membranes, evidencing a preference for the latter. Moreover, we determined their antifungal, hemolytic, and cytotoxic activities. Specifically, we proved the antifungal activity against a relevant panel of pathogenic fungi belonging to *Candida* spp., and biofilms formed by *C. albicans*, *C. tropicalis*, and *C. auris*. Finally, the more active lipopeptide was used in a pilot study on an in vivo model consisting of *Galleria mellonella* larvae infected with *C. auris.* The tested lipopeptide was effective in counteracting infection and death of the larvae.

## 2. Results and Discussion

### 2.1. Peptide Design and Physico-Chemical Properties

The first step in the activity and selectivity of cationic AMPs is the electrostatic interaction with the negatively charged bacterial membrane [23,24,25]. Thus, different strategies are used to improve the antibacterial activity with the aim to create more efficient potential agents in clinical. The most-used approach is to find the correct balance between the net positive charge and the amphipathic properties to obtain peptides that can fold in α-helices or β-structures [22,26,27,28]. In the present study, we designed small peptides derived from the sequence of KHS-Cnd [22]. KHS-Cnd was chosen because of its high antimicrobial activity against Gram-positive, Gram-negative, and multidrug-resistant bacteria and relatively low cytotoxicity. We synthesized three small peptides of 11 amino acids from the parental peptide. The sequences and physicochemical properties of the three derivatives corresponding to the N-terminal, central, and C-terminal portions of KHS-Cnd, named Pep-A, Pep-B, and Pep-C, are reported in Table 1. Many studies report that the functionalization with fatty acid chains increases antifungal activity [20,21]. Thus, we modified the N-terminal by introducing a hydrophobic myristoyl moiety (14-carbon fatty acid) to favor the vehiculation and interaction of peptides with membranes and enhance their membrane-anchoring properties [29,30]. This modification increased the hydrophobicity of peptides without varying the peptide sequence.

Physico-chemical properties of myristoylated peptides, called Myr-A, Myr-B, and Myr-C, are also reported in Table 1.

### 2.2. Critical Micellar Concentration (CMC) Determination

Lipopeptides usually self-assemble in organized structures that have attracted interest because of their enhanced bactericidal and fungicidal effects [31,32]. The self-assembly of myristoylated peptides was evaluated by determining the critical micellar concentration (CMC) that represents the concentration of spontaneous associations of monomers by a spectrofluorometric method. 8-anilinonaphtalene-1-sulfonate (ANS) was used as a fluorescent probe since its fluorescence is very sensitive to the microenvironment, showing a blue shift and intensity increase upon changing from a polar to non-polar medium [33]. Therefore, the fluorescence of ANS was monitored as a function of the lipopeptide concentration. Figure 1A shows the emission spectra of ANS at increasing concentrations of myristoylated peptides (Myr-C). Figure 1B shows the plot of fluorescence intensity of ANS (the maximum emission wavelength was 465 nm) vs. the logarithm of the lipidated peptide concentration. Fluorescence intensity slowly increases up to a certain lipopeptide concentration and then breaks abruptly, indicating a sudden change in the aggregation state of lipopeptides. For each peptide, the critical micellar concentration (CMC) was evaluated by the intersection of the two lines. The values of CMC are 4.7 ± 0.8 μM, 5.1 ± 0.8 μM, and 5.1 ± 0.4 μM for Myr-A, Myr-B, and Myr-C, respectively (data for Myr-A and Myr-B are reported in Appendix A).

Non-lipidated peptides did not show a break in the fluorescence intensity upon titration, suggesting that they do not aggregate.

### 2.3. Membrane Partition Studies

To quantify the interaction of different lipidated and non-lipidated peptides with membrane mimetic systems, we measured the intrinsic fluorescence of Trp-1 upon partition with LUVs (large unilamellar vesicles) of different compositions. Briefly, the mixtures of POPC/POPG (70%/30% *w*/*w*) LUVs were used to mimic anionic membranes and POPC only to mimic zwitterionic membranes [34,35,36]. LUVs do not entirely describe the complexity of natural membranes; still, they are used here to evaluate the preference of different peptides for anionic (bacteria and fungi) and zwitterionic (mammalian cell) membranes. The mole fraction partition constants *K_x_* were calculated from the partition isotherms as previously described [37]. Figure 2 reports the partition isotherms for the titration of myristoylated and non-myristoylated peptides with POPC and POPC/POPG vesicles.

The values of *K_x_* for all peptides are reported in Table 2 together with Δ*G* values, calculated using the expression Δ*G = −RT lnK_x_*, and the selectivity ratio defined as the ratio between partition constants determined in anionic and zwitterionic membranes. Higher values of the selectivity ratio represent a preference of the peptide for anionic vesicles, while lower values indicate a preference for the zwitterionic one.

Partitioning data suggest that myristoylated peptides interact with LUVs with higher affinity. The mole fraction partition coefficients *K_x_* range from 3.0 × 10^4^ to 5.2 × 10^4^ with calculated Gibbs free energies of the partition of ~(−26) kJ/mol for non-myristoylated peptides, suggesting poor binding to membranes. Myristoylated peptides showed larger *K_x_* values ranging from 1.0 × 10^5^ in POPC to 1.7 × 10^6^ POPC/POPG vesicles, indicating a stronger interaction with LUVs. 

Moreover, myristoylated peptides prefer to interact with anionic membranes as described by the values of the selectivity ratio ranging between ~2 and 4. This feature should increase the antifungal activity, as described later. Mc Laughlin and colleagues stated that the presence of the myristoyl moiety is required to interact with membranes but is not always sufficient [38]. The higher propensity of myristoylated peptides to interact with membranes can be explained by the presence of the myristoyl group and charge effects. Mainly, the values of Gibbs free energy ~(−33) kJ/mol are similar to that measured for the insertion of myristate into the bilayer [39]. Thus, the synergism between the myristate insertion into the bilayer and the electrostatic interaction between the lysine residues present in the peptides and the anionic lipids can be the driving force of the process [40].

### 2.4. Fluorescence Quenching Experiments

Peptides’ topology in the absence and presence of lipid vesicles has been evaluated by monitoring the quenching of Trp-1 by acrylamide. Acrylamide is a small neutral molecule that quenches the tryptophan residue with high efficiency [41]. Quenching experiments were carried out in buffer and lipid vesicles as previously described [42]. Typical results are shown in Figure 3 for myristoylated and non-myristoylated peptides.

In the buffer, all non-myristoylated peptides and myristoylated peptides Myr-A and Myr-C showed typical linear behavior of the fluorescence intensity ratio in the absence (*F_o_*) and in the presence (*F*) of the quencher vs. the quencher concentration. The same behavior was detected for myristoylated peptides in POPC and POPC/POPG lipid vesicles. The classical Stern–Volmer equation [43]
(1)$$\frac{F_{o}}{F}=1+K_{SV}\left[Q\right]$$
was used to describe the fluorescence quenching, where *K_SV_* is the Stern–Volmer or collisional constant, while *F_o_* and *F* are the fluorescence in the absence and presence of the acrylamide Q, respectively. Values of *K_SV_* are indicative of the dynamic quenching and are higher in buffer solution than in the presence of lipid vesicles (Table 3), suggesting that Trp-1 is less accessible to the quencher in the presence of lipid vesicles and confirming the partition into the lipid bilayer.

As reported in Figure 3, Myr-B in the buffer and non-myristoylated peptides in the presence of lipid vesicles showed non-linear behavior, with downward curvature at high quencher concentrations. This effect can be ascribed to selective quenching [44] due to the presence of two different fluorophore populations (*a* and *b*) with a diverse solvent exposition: one more accessible to the quencher and the other less. The fluorescence intensity can be written as:(2)$$F_{o}=F_{o,a\;}+F_{o,b}$$

The fraction of total fluorescence has been determined using the following equation [43]:(3)$$\frac{F_{o}}{F-F_{o}}=\frac{1}{f_{a}}+\frac{1}{f_{a}K_{a}\left[Q\right]}$$
where *f_a_* is the fraction of the accessible fluorophore at an infinite concentration of the quencher and *K_a_* is an average Stern–Volmer constant for the accessible fluorophore both exposed and buried. A plot of, $\frac{F_{o}}{F-F_{o}}\;vs.\;\frac{1}{\left[Q\right]}$ yields a straight line from which we determined the fitting parameters *f_a_* and *K_a_* (Appendix A). Values of *f_a_* for non-myristoylated peptides in the presence of lipid vesicles are between 0.40 and 0.75, indicating two fluorophore populations, one accessible to the quencher and the other deeply buried in the phospholipid bilayer.

Furthermore, we calculated the NAF, defined as the ratio between the Stern–Volmer constant in the presence and absence of LUV [45]:(4)$$\mathrm{NAF}=\frac{{\left(K_{SV}\right)}_{\mathrm{LUV}}}{{\left(K_{SV}\right)}_{\mathrm{Buffer}}}$$

Low values of NAF suggest protection of the fluorophore from the quencher and strong interaction with lipids. NAF values reported in Table 3 are lower for myristoylated peptides than for non-myristoylated, suggesting a better interaction with the lipid vesicles. Peptide Myr-B has a lower value of NAF (0.07), indicating a strong interaction with the lipid and a preference for anionic lipid vesicles. Non-myristoylated peptides show high values of NAF, suggesting a weak interaction with lipid vesicles. All the data agree with the results obtained from the partition studies.

### 2.5. Circular Dichroism Experiments

Structural transitions of peptides upon the addition of lipid vesicles have been investigated by CD spectroscopy [42]. We explored structural changes of both myristoylated and non-myristoylated peptides upon interaction with anionic and zwitterionic membrane-mimicking systems. In Figure 4, typical titration curves for myristoylated peptides are reported at increasing lipid vesicle concentrations. The explored peptide-to-lipid ratios ranged from 1:0 to 1:40. CD data confirm that all peptides can interact with lipid vesicles of different compositions. Peptides Myr-A and Myr-C show similar CD spectra in buffer (black curves) with a minimum at ~220 nm, typical of a β-conformation. This conformation seems to be stable, and it also persists upon the addition of POPC lipid vesicles. However, there is a slight transition towards a partially helical conformation in the presence of anionic lipid vesicles. The peptide Myr-B shows a typical spectrum of a random coil in buffer and in the presence of lipid vesicles of different compositions.

The fractional helical content *f_α_* for the peptides Myr-A and Myr-C in the presence of POPC/POPG lipid vesicles has been estimated using the formula [46]:(5)$$f_{\alpha }=\frac{\theta -\theta_{RC}}{\theta_{H}-\theta_{RC}}\;$$
where θ is the observed ellipticity at 222 nm and $\theta_{RC}$ and $\theta_{H}$ are the values for a completely random coil and an α-helical conformation, respectively. According to the Luo and Baldwin [47] formula, the limiting values are $\theta_{RC}=-360$ and $\theta_{H}=-39.375$. The helical fractions *f_α_* in the presence of POPC/POPG lipid vesicles are ~0.2 and ~0.1 at an L/P molar ratio of 40 for Myr-A and Myr-C, respectively.

### 2.6. Molecular Dynamics Simulations

To investigate more deeply, with an atomic resolution, the interaction between the peptides and the membrane models, we carried out a series of molecular dynamics simulations of peptides in different environments, including water, TFE/water (50% *v*/*v*), and in the presence of POPC and POPC/POPG lipid bilayers.

The initial structure of peptides used for the MD in the solution was obtained using the I-Tasser bioinformatics tool, which suggested, for all peptides, an α-helix conformation.

The MD simulations in water of all peptides showed a rapid loss of the initial α-helix conformation (Appendix A). Only for Myr-B was the formation of a transient α-helix observed during 300 ns of simulation. Analysis of the peptide’s secondary structure as a function of the simulation time suggests that all peptides are unstructured with high conformational flexibility in the water, in agreement with the experimental results obtained by CD spectra.

In the presence of TFE/water (50% *v*/*v*), a less polar solvent with respect to water, Pep-B, Pep-C, and Myr-C conserved the α-helix conformation for 300 ns with a helicity of about 45%, 64%, and 82%, respectively (Appendix A). In the case of Pep-A and Myr-A, the analysis of the secondary structure showed an unfolding of peptides after 120 ns and 220 ns of simulation (Appendix A).

We also simulated the adsorption of peptides on the surface of anionic and zwitterionic membrane models. The structures obtained at the end of the MD simulations in TFE/water were used as the initial conformation of peptides for the simulations in the presence of POPC and POPC/POPG lipid bilayers. The peptides were positioned close to the surface of the membranes with the principal axis parallel to the lipid surface and a distance of 1.0 nm between the center of mass of the peptide and the phosphorous atoms of lipids.

Figure 5 shows the representative snapshots of MD simulation at 1 μs of myristoylated peptides in the presence of POPC and POPC/POPG lipid bilayers. Myr-A and Myr-B interact with the surface of the zwitterionic and anionic membrane through an unfolded conformation, while Myr-C shows a higher tendency with respect to the other myristoylated peptide to fold in the presence of lipid bilayers. In particular, analysis of the time evolution of the secondary structure of Myr-C in POPC/POPG shows that the peptide conserves helicity (α-helix and π-helix) of about 45% (five residues) during the simulation duration (Appendix A). On the other hand, in the presence of POPC, Myr-C interacts with the membrane, adopting, in the last 500 ns of simulation, a 3_10_-helix with helicity of about 45.5% (Appendix A).

The adsorption of myristoylated peptides to the surface of the lipid bilayer is driven mainly by the alkyl chain, linked to the N-terminal residue of peptides, and by the presence of positively charged amino acids. The alkyl chain penetrates in the hydrophobic region of the lipid bilayer while the positively charged amino acids stabilize the peptide–membrane interaction by forming hydrogen bonds and electrostatic interactions with lipid molecules. The peptide Myr-A in POPC shows a deep penetration of the myristoyl chain and Trp-1 in the hydrophobic region of the membrane (Figure 5A and Appendix A). In particular, the average position of Trp-1 is located slightly below the position of the carbonyl group (CO*_sn_*_-1_ and CO*_sn_*_-2_) of POPC, as evidenced by the analysis of the average density distribution calculated to the lipid bilayer center (Appendix A). In POPC/POPG (Figure 5B and Appendix A), the residues of Trp-1 and Lys-4 are located close to the lipid/water surface. The residue Arg-7 interacts with phosphate groups forming an arginine-phosphate salt bridge. The different penetrations into the lipid bilayer of POPC and POPC/POPG observed by MD simulation for Trp-1 of the peptide Myr-A is in agreement with the NAF values obtained by fluorescence-quenching experiments. In POPC/POPG, Trp-1 is close to the lipid surface and then more exposed to the quencher molecules with respect to POPC, where the aromatic residue can penetrate the hydrophobic region of the membrane.

The peptides Myr-B and Myr-C adsorbed on the surface of the lipid membrane show a different orientation with respect to Myr-A. The peptide Myr-B, in the presence of POPC and POPC/POPG, adopts an extended conformation parallel to the lipid surface with charged residues that interact with phosphate groups of lipids, and the residue Trp-1 located in the hydrophobic region of the membrane (Figure 5C,D). Myr-C, which adopts a folding conformation (5–6 residues) when adsorbed by the membrane, in the presence of POPC shows residues of Trp-1 and all lysine residues except Lys-11 located inside the membrane close to the carbonyl group of lipids (Figure 5E and Appendix A). On the contrary, in POPC/POPG, all charged amino acids of the peptide Myr-C are located close to the phosphate group (Figure 5F and Appendix A) and interact with lipid molecules via the salt bridge and hydrogen bonds. In the case of Myr-B and Myr-C, analysis of the density distribution calculated with respect to the lipid bilayer center suggests that in POPC/POPG, the residue of Trp-1 is located more inside the membrane with respect to POPC, in agreement with the factor NAF obtained by fluorescence experiments.

The analysis of MD trajectories of non-myristoylated peptides in POPC and POPC/POPG reveals a lower affinity of peptides for lipid bilayers with respect to myristoylated peptides, as shown by the partition constants *K_x_* reported in Table 2. In particular, the peptides Pep-B and Pep-C are adsorbed on the lipid surface and cannot deeply penetrate the membrane but lie on the surface, with the non-charged amino acids oriented toward the water phase (Appendix A). Figure 6 shows, for example, the different penetrations of Pep-C and Myr-C adsorbed on the lipid surface of POPC/POPG.

The density distribution of Trp-1 atoms, calculated with respect to the lipid bilayer center (Appendix A), shows that for all non-myristoylated peptides, Trp-1 does not penetrate the hydrophobic portion of the membrane but remains located close to phosphate groups of lipids at the lipid/water interface. At the lipid/water interface, the residue Trp-1 is more exposed to water molecules and all molecular species present in the water phase. MD simulation results suggest that Trp-1 of myristoylated peptides is deeply immersed in the hydrophobic region of the membrane and, therefore, poorly exposed to water, with respect to non-myristoylated peptides. These results are in good agreement with the fluorescence experiments.

The MD simulations evidenced that the presence of the myristoyl moiety linked to Pep-A, Pep-B, and Pep-C deeply influences the properties of peptides, particularly their structure and affinity for lipid bilayers.

### 2.7. Antifungal Activity

The MIC_90_ values of non-myristoylated and myristoylated peptides against different *Candida* spp. are summarized in Table 4. All lipopeptides have much lower MIC_90_ values than their unfunctionalized counterparts, with the sole exception of the Myr-A peptide that maintains an unchanged MIC_90_ value towards *C. auris*. As already observed in the literature, these results confirm that increasing hydrophobicity by fatty acylation of AMPs with myristic acid enhances their antimicrobial activity and, in particular, the antifungal properties [48]. The evidenced MIC_90_ values are between 5 and 21 µM for all selected *Candida* species, while only *Candida* auris showed a higher range with Myr-B, which is the most active (MIC_90_ 20 µM). Myr-A and Myr-C were the most active against the selected *Candida* species, and Myr-B particularly showed a MIC of 8 μg/mL (10 μM) vs. *C. tropicalis*. Values of the single MICs of the 10 isolates for *Candida species* are reported in the Appendix A. MIC values are higher than CMC, so MIC values are determined in the presence of lipopetide micellar aggregates and not the free monomer. Under these conditions, lipopetides are still soluble even if in the form of micellar aggregates. Thus, the activity of lipopetides might also depend on the so-called micellar mechanism, consisting of the solubilization of part of the host membrane into mixed micelles. However, further studies are needed to verify this hypothesis.

### 2.8. SEM Analysis

The antibiofilm activity of the Myr-B lipopeptide was tested on representative high-biofilm-producing strains of *C. albicans*, *C. tropicalis,* and *C. auris*. These strains were selected for their ability to form biofilms in vitro as assessed by the crystal violet assay (data not shown). SEM images clearly show that Myr-B is able to demolish the biofilm of all three fungal isolates at a concentration of 128 μg/mL (85 μM) (Figure 7). Untreated biofilms show a well-organized three-dimensional structure, with the presence of an extracellular matrix for the *C. albicans* and *C. tropicalis* strains and fungal hyphae, well-branched and adhered to the surface. The biofilm of *C. auris*, as already reported [49], is characterized by cellular aggregates with a poor extracellular matrix able to tolerate clinical concentrations of sodium hypochlorite. After treatment with the Myr-B lipopeptide, the biofilm of all fungal strains is strongly demolished. Few cellular hyphae fragments can be observed, but the morphological organization is completely lost. Biofilm demolition results validate the idea the peptide’s myristoylation strongly enhances the antifungal properties also towards the biofilm form. These results have a significant impact on the search for new peptide drugs able to combat the rising antimicrobial resistance and also effective against biofilm-related infections. 

### 2.9. In Vivo Toxicity and Efficacy Testing in Galleria Mellonella Model

Multidrug-resistant *C. auris* was chosen as a representative species for in vivo studies. After determining the infective fungal dose of 10^7^ CFU/injection was capable of killing about 50% of the *Galleria mellonella* larvae 24 h after infection (data not shown), we evaluated the in vivo toxicity and efficacy of the Pep-B and Myr-B peptides (the Myr-B peptide was selected as the best active peptide due to the previously shown results). Groups of ten *G. mellonella* larvae were infected with *C. auris* and treated with the Pep-B and Myr-B peptides at a concentration of 640 µg/mL (about 400 µM). The results shown in Table 5 highlighted that the toxicity of Pep-B is slightly higher than that of the corresponding lipopeptide. Furthermore, Myr-B was able to contain the effects of fungal infection 24 h after the treatment, with a single addition of Myr-B: 8 out of 10 larvae survived, whereas, in the control group, only 4 survived. In the group of infected larvae treated with Pep-B, 7 out of 10 larvae died after 24 h, showing poor efficacy compared to the control. In these pilot and preliminary studies with a single drug administration, the data obtained after 24 h are the most significant, as in the following 48 h, the injected compound undergoes inactivation and elimination likely due to the action of internal proteases. Therefore, we could also confirm the antifungal activity of the Myr-B peptide in vivo. 

### 2.10. Cytotoxicity Assay against Human Cell Line

The cytotoxic effect of the six peptides has been investigated on a primary human fibroblast cell line (FB789). Four different peptide concentrations (5, 10, 20, and 50 µM) and two different time points (12 and 24 h) have been tested (see Figure 8). Pep-A, Pep-B, and Pep-C [see panel A–C] and the corresponding Myr-A, Myr-B, and Myr-C [see panel D–F] evidenced no toxicity even at the highest peptide concentration. Moreover, Pep-A, and to a lesser extent Myr-A, showed a proliferative effect. No cytotoxic effect was found in the peptide Myr-B concentrations that were active, both in vitro and in vivo, against the tested fungal species.

### 2.11. Hemolytic Assay against Rabbit Erythrocytes

Rabbit erythrocytes have been used to investigate the hemolytic effect of the six peptides (Figure 9). Hemolysis values from 5 to 20 µM are rather low, ranging from 0.1% (Peptide-A and Peptide Myr-A at 5 µM, see panel A) to 7% (Peptide Myr-B, see panel B). An increase in the membrane lysis induced by the peptides is visible at 50 µM, with an effect of 28.5% of hemolysis reached by Peptide Myr-C (see panel C). At the concentrations of Myr-B active against *Candida* species, no or very few hemolytic effects have been evidenced. 

## 3. Conclusions

In recent years, fungal infections have been increasing worldwide, and they are usually associated with relatively high mortality rates. Still, the current antimycotic therapies are limited. Moreover, some *Candida* species resistant to antibiotics have emerged and new antifungal drugs need to be developed [50]. In this paper, we reported the design and characterization of three lipopeptides active in vitro against different multidrug-resistant *Candida* species. Lipopeptides were selective and active against fungal cells and not harmful against either fibroblast mammalian cells or erythrocytes. Finally, as the first proof of evidence, we tested the in vivo activity of the most-active peptide, Myr-B, against an infection of *Candida auris* in insect larvae of *Galleria mellonella*. The peptide was not toxic and could contain the effects of fungal infection 24 h after the treatment. Therefore, it could be considered a new promising drug candidate. These results could greatly impact the search for new antifungal peptide drugs able to combat the rising antibiotic resistance and against biofilm-related infections.

## 4. Materials and Methods

### 4.1. Materials

All the reagents were purchased from Sigma-Merck (Milan, Italy) unless otherwise specified. All peptides and lipo-peptides were purchased from CASLO ApS, c/o Scion Technical University of Denmark, with a purity >98%. Peptide concentrations were determined by UV spectroscopy at 280 nm before each sample preparation.

### 4.2. Lipid Vesicles Preparation

The lipids POPC (1-palmitoyl-2-oleoyl-*sn*-glycero-3-phosphocholine) and POPG (1-palmitoyl-2-oleoyl-*sn*-glycero-3-phosphoglycerol) were purchased from Avanti Polar Lipids (Alabaster, AL, USA) and used without further purification. LUVs (Large Unilamellar Vesicles) were prepared as previously reported [37]. Briefly, appropriate amounts of lipids were dissolved in chloroform/methanol 9:1. The solvent was then removed by rotary evaporation and samples were placed overnight under a high vacuum. The lipid film was then hydrated by adding 1 mL of buffer (20 mM phosphate buffer at pH 7.4 with 150 mM NaCl and 0.8 mM EDTA) and then subjected to 5 freeze–thaw cycles and vortexed. The multilamellar vesicles (MLVS) were extruded through a polycarbonate membrane with pores of 100 nm by an Avanti polar mini extruder, and the obtained LUVs were used within 48 h of preparation. LUVs were composed of 100% POPC and a 70%/30% (*w*/*w*) combination of POPC/POPG. The final concentration of the lipid was 10 mM.

### 4.3. Fluorescence Spectroscopy

All the steady-state fluorescence measurements were carried out using a Perkin Elmer LS55 fluorescence spectrophotometer. All the experiments were conducted at 298 K in a thermostatic cell holder and in 20 mM phosphate buffer at pH 7.4 containing 0.8 mM EDTA and 150 mM NaCl unless otherwise specified.

#### 4.3.1. Critical Micellar Concentration (CMC) Determination

The CMC of the N-terminal myristoylated peptides was determined in water by fluorescence spectroscopy using 8-anilinonaphtalene-1-sulfonate (ANS) as the fluorescent probe [51]. The ANS assays were performed using peptide concentrations of 0.8–10.0 μM in 70 μM ANS solution.

The ANS was excited at λ_ex_ = 356 nm, and the fluorescence spectrum was measured for λ_em_ = 400–670 nm with excitation and emission bandwidths of 5 nm.

#### 4.3.2. Partition Constant Determination

Since all peptides contain tryptophan, fluorescence spectroscopy is an appropriate tool for studying these molecules. Peptide partitioning between buffer and lipid vesicles was monitored by measuring the increase in tryptophan fluorescence upon the addition of LUVs. Trp fluorescence spectra were recorded with λ_exc_ = 295 nm and scanning the emission between 305 and 400 nm. Measurements were carried out with a cross-oriented configuration of polarizers (pol_em_ = 0° and pol_exc_ = 90°) [52]. Briefly, a 1.0 μM solution of the peptide was titrated with LUVs of different compositions (POPC 100% and 70%/30% POPC/POPG) as previously described [22,37]. The background effects of vesicles and buffer were subtracted from each spectrum. All the fluorescence emissions were corrected for dilution. Mole fraction partition coefficients *K_x_* were calculated from the fraction of the partitioned peptide *f_p_* as previously described [22,37]. The fraction of the partitioned peptide *f_p_* is given by the following equation [53]
(6)$$f_{p}=\frac{K_{x}\left[L\right]}{\left[W\right]+K_{x}\left[L\right]}$$
where *K_x_* is the mole fraction partition constant, and [*L*] and [*W*] are the molar concentrations of lipids and water, respectively.

#### 4.3.3. Acrylamide Quenching Experiments

Quenching of tryptophan fluorescence both in the absence and presence of lipid vesicles was accomplished by the addition of small aliquots of acrylamide dissolved in buffer. The concentration was 2.0 μM for both myristoylated and non-myristoylated peptides whereas lipid vesicles were 400 μM in the experiments with myristoylated peptides and 1.0 mM in the experiments with non-myristoylated peptides. The excitation wavelength was set at 295 nm, and fluorescence emission spectra were recorded between 305 and 400 nm. Measured fluorescence was corrected for dilution of the sample, and data were fitted according to the Stern–Volmer or modified Stern–Volmer equation [43]. 

### 4.4. Circular Dichroism Spectroscopy

Circular Dichroism (CD) spectra were recorded in the 190–260 nm spectral range at 298 K on a J 715 JASCO spectropolarimeter with 1.0 nm intervals, using a 0.2 cm path-length quartz cuvette. The reported spectra are the average of 8 scans with an instrument scanning speed of 100 nm/min, a step size of 1 nm, and 1s response. Liposomes for CD experiments were prepared as described in Section 4.2 using a 10 mM phosphate buffer (pH 7.4 and 0.1 mM EDTA).

CD spectra of peptides were recorded in 10 mM phosphate buffer (pH 7.4 and 0.1 mM EDTA) in the absence and presence of LUVs of different compositions. The general procedure for non-myristoylated peptides was as follows: To a freshly prepared solution of 30 μM peptide, increasing aliquots of the proper liposome solution were added. After each addition, the solution was left for 5 min to equilibrate under stirring before recording the CD spectrum. The explored lipid/peptide ratios ranged from 1:0 to 1:60.

For myristoylated peptides, six different samples at a 30 μM lipopeptide concentration and increasing liposome concentrations (30 μM, 150 μM, 300 μM, 600 μM, 900 μM, and 1200 μM) were prepared. The samples were allowed to equilibrate for 1 h before recording the CD spectrum.

The CD signals in millidegrees *(θ*) were converted into mean molar ellipticities (deg cm^2^ dmol^−1^) and plotted vs. the wavelength.

### 4.5. Molecular Dynamics Simulations

#### 4.5.1. Simulation Systems

The initial structure of Pep-A, Pep-B, and Pep-C were obtained using the fold recognition algorithm implemented in I-TASSER (iterative threading assembly refinement algorithm) [54]. Arginine and lysine side chains were considered positively charged according to their protonation state at pH = 7. The structure of myristoylated peptides, Myr-A Myr-B, and Myr-C, were obtained by linking, with an amide bond, the acyl chain to the N-terminal tryptophan residue of Pep-A, Pep-B, and Pep-C using Avogadro software [55].

Four systems were constructed for each peptide. In the first system, the peptide was immersed in a cubic box 5.0 × 5.0 × 5.0 nm^3^ and solvated with at list 4000 water molecules. In the second system, the peptide was immersed in a cubic box 5.0 × 5.0 × 5.0 nm^3^ and solvated with a mixture of TFE/water (50% *v*/*v*). In the third system, the peptide was positioned close to the surface of the pre-equilibrated bilayer of POPC with the principal axis parallel to the lipid surface and a distance of 1.0 nm between the center of mass of the peptide and the phosphate atoms of lipids. A rectangular box of 6.5 × 6.5 × 9.0 nm^3^ was used with at least 6500 water molecules. In the fourth system, each peptide was positioned above the pre-equilibrated bilayer surface of POPC/POPG with the principal axis parallel to the lipid surface and a distance of 1.0 nm between the center of mass of the peptide and the phosphate atoms of lipids. A rectangular box of 6.2 × 6.2 × 9.0 nm^3^ was used with at least 6500 water molecules.

For the simulation in water and the mixture of TFE/water, the best-scored structures of each peptide obtained by the I-TASSER prediction were used. For simulations in the presence of the lipid bilayer of POPC and POPC/POPG, the structure of the peptide obtained at the end of the MD simulation in TFE/water was used to prepare the initial configuration of the system with the peptide close to the surface of the bilayer. The initial configuration of POPC and POPC/POPG (70:30) was built with 128 lipids (64 per leaflet). The mixed bilayer of POPC/POPG (70:30) was built using 92 molecules of POPC and 36 molecules of POPG (18 per leaflet).

#### 4.5.2. Simulation Details

All the molecular dynamics simulations were performed using the GROMACS 2020.3 packages [56]. The GROMOS 54A7 force field [57] was used to model the peptides’ molecules and ions in conjunction with the GROMOS 54A8 containing the new lipid head group parameters proposed by Marzuoli et al. [58] for the lipid molecules of 1-palmitoyl-2-oleoyl-*sn*-glycero-3-phoshocholine (POPC), 1-palmitoyl-2-oleoyl-*sn*-glycero-3-phospho-L-(1-glycerol) (L-POPG), and 1-palmitoyl-2-oleoyl-*sn*-glycero-3-phospho-D-(1-glycerol) (D-POPG). The phosphatidylglycerol component of the mixed lipid bilayer was a racemic mixture of L-POPG and D-POPG. Water was modelled with the SPC model [59], and the TFE molecules were described using the force field of Fioroni et al. [60].

Van der Waals and electrostatic interactions were calculated using a cut-off of 1.2 nm, and the particle mesh Ewald method (PME) [61] was applied to the long-range electrostatic interactions beyond 1.2 nm. Covalent bonds were constrained using the P-LINCS algorithm [62,63], whereas the geometry of water was constrained with the SETTLE algorithm [64]. Newton’s equation of motion was integrated via the leap-frog algorithm using a time step of 2 fs.

The velocity-rescale thermostat [65] was used to control the temperature of the simulated systems by coupling the peptide, lipids, water, and ions separately to an external bath at 298 K with a coupling constant $\tau_{T}$ of 0.1 ps. The pressure was controlled using the Berendsen barostat [66] (P = 1 bar, τ_p_ = 1.0 ps). The pressure coupling was isotropic in the case of the simulations in water and in the mixture of TFE/water (50% *v*/*v*) and semi-isotropic in the presence of lipid bilayers. Periodic boundary conditions were applied in all three dimensions.

The trajectories obtained by MD simulation were analyzed with the GROMACS analysis tools, VMD 1.9.3 [67], and in-house scripts. Analysis of the secondary structure was performed with the gmx do_dssp tool of GROMACS using the program DSSP [68].

### 4.6. Antifungal Activity of Peptides

#### 4.6.1. Fungal Strain, Media, and Culture Conditions

The antifungal activity of AMPs was determined against 50 clinical isolates, 10 for each of the following Candida species: *C. albicans*, *C. tropicalis*, *C. glabrata*, *C. parapsilosis*, and *C. auris*. The latter were derived from patients with positive blood cultures in four hospitals in India, with known resistance profiles to fluconazole, voriconazole, and echinocandins. Yeast cells were grown in Brain Heart Infusion (BHI) medium (Sigma-Aldrich, Gillingham, UK) for 16 h at 37 °C in a 150-rpm orbital shaker. Cells were then sub-inoculated in a fresh BHI medium and grown to an optical density (OD) of 0.3. The turbidity of the inoculum was adjusted to 0.5 McFarland and diluted 1:500 in RPMI 1640 broth, corresponding to around 2.5 × 10^5^ CFU/mL.

#### 4.6.2. MIC Evaluation

Minimal inhibitory concentrations (MICs) were determined by Broth microdilution susceptibility tests, according to The Clinical and Laboratory Standards Institute (CLSI) international guidelines [69]. A fungal suspension of 0.5 MacFarland was diluted 1:500 in RPMI 1640 broth and incubated in 96 multi-well plates containing AMPs at different concentrations ranging from 0.125 to 256 μg/mL. In this range of concentration, lipopeptides were completely soluble. MIC was defined as the lowest concentration of the peptides at which yeasts showed no visible growth compared to the control. The MIC_90_ represents the value at which ≥90% of the strains within a test population were inhibited.

#### 4.6.3. Biofilm Conditions and Biofilm Demolition Assays

Fungal strains of *C. albicans*, *C. tropicalis,* and *C. auris* were cultured in BHI and grown at 37 °C overnight, harvested by centrifugation (3000× *g*, 10 min), resuspended in sterile PBS, and adjusted to a density of 10^9^ colony-forming units (CFU)/mL in RPMI media by measuring the absorbance at 600 nm. A 24-well plate with 13-mm-diameter glass coverslips (Thermanox NUNC, New York, NY, USA) was seeded with 500 mL and grown 3 days at 37 °C for biofilm formation. Once the biofilms were formed, after 3 days incubation, the culture medium was removed, and cells were washed three times with PBS to remove non-adherent cells. The fungal biofilms, highly visible on the disk surface, were then treated with a 200 µL final volume of peptide X in PBS at two different concentrations (128 and 16 µg/mL) for 16 h at 37 °C. The demolition activity was observed in scanning electron microscopy (SEM).

#### 4.6.4. SEM Analysis

Fungal biofilms were analyzed by SEM (Supra 25-Zeiss, Carl Zeiss AG, Oberkochen, Germany). Fixed and dried biofilms were mounted onto an aluminum stab using double-sided carbon tape and coated with a gold/palladium film (80:20) using a high-resolution sputter coater (SCD-040, Balzers Union Ltd., Balzers, Lichtenstein).

#### 4.6.5. Galleria Mellonella Model

Ten larvae (250–320 mg each) were selected at random for each step in the procedure. Any larvae with darkening of the cuticle were discarded. The test compounds and yeast were injected into the hemocoel in water buffer through the last left proleg (Insumed 30G Pic, Italy, syringe volume 0.5 mL, needle size 30G) [70]. The larvae were incubated in the dark for 3 days, and mortality was recorded daily.

An infective dose of yeast *Candida auris* was determined by injecting groups of ten larvae with a suspension at 10^5^, 10^6^, 10^7^, and 10^8^ colony-forming units (CFU) per injection. The larvae were incubated for 3 days at 30 °C. An infective dose was defined as the one that caused an immune response, recognizable by the darkening of the cuticle and able to kill approximately 50% of larvae after 24 h [70].

For the peptide efficacy experiment, ten larvae were injected into the last left proleg with a pre-determined infective dose of yeast and the dose of peptides at the same time in a final volume of 10 μL and incubated at 30 °C. Each step in the procedure included six groups of 10 larvae: *C. auris*-infected larvae (10^7^ CFU), infected and treated larvae with peptide B (640 μg/mL), infected and treated larvae with peptide Myr-B (640 μg/mL), treated larvae with peptide B (640 μg/mL), treated larvae with peptide Myr-B (640 μg/mL), and vehicle-injected control (water). The mortality was recorded daily for 3 days.

### 4.7. Cytotoxicity Assay against Human Cell Line

The cytotoxicity of the six peptides was tested on a primary human fibroblast cell line (FB789) grown in Dulbecco’s Modified Eagle Medium (DMEM) containing 10% fetal calf serum (FCS) and antibiotics (penicillin–streptomycin, Gibco) at 37 °C in a humidified 5% CO_2_ atmosphere. Cells were seeded on 96-well microplates at a density of 3 × 10^3^ cells per well in 100 µL of the medium. Four dilutions of the six peptides (from 5 µM to 50 µM) were added to the cells and maintained for 12 and 24 h. As a negative control, cells were grown in a normal medium without the peptides, while, as a positive control, cells were added to NaN_3_ 10% *v*/*v*. The cytotoxicity was determined by measuring the intracellular adenosine triphosphate (ATP) levels using the luciferase-based ATPlite assay (PerkinElmer, Waltham, MA, USA), according to the manufacturer’s instructions. After 12 and 24 h, the cells were lysed, and the lysates were transferred into opaque well plates (OptiPlate-96, PerkinElmer). The amount of emitted light, linearly correlated with the ATP concentration, was measured with a microplate luminometer (Victor II PerkinElmer) for 10 min in the dark. Three replicates for each dilution were performed. Cell viability values were expressed as the mean + SD and calculated as the percent values of the treated samples with respect to the untreated cells (negative control). The statistical analysis was performed using one-way ANOVA analysis followed by the Dunnett post-test.

### 4.8. Hemolytic Activity Assay

The hemolytic assay was performed on rabbit erythrocytes (Rockland) maintained in Alsever’s solution (Innovative Research). Briefly, after the removal of Alsever’s solution, the erythrocytes were resuspended in PBS and counted in a hemocytometer. A suspension of 5,000,000 red blood cells was incubated with four dilutions (from 5 µm to 50 µm) of the six peptides in a 96-well microplate. As a negative control, erythrocytes were incubated without the peptides, while as a positive control, erythrocyte was added with triton 10% *v*/*v*. The plate was incubated at 37 °C for 2 h and subsequently centrifuged at 1200 rpm × 3 min to separate the pellet from the supernatant. The absorbance of the supernatant was measured at 492 nm. Each point was made in triplicate. The relative OD compared to the positive control defined the percentage of hemolysis [71]. Statistical analysis was performed using two-way ANOVA analysis followed by Bonferroni’s post-test.

## Figures and Tables

**Figure 1 ijms-23-02164-f001:**
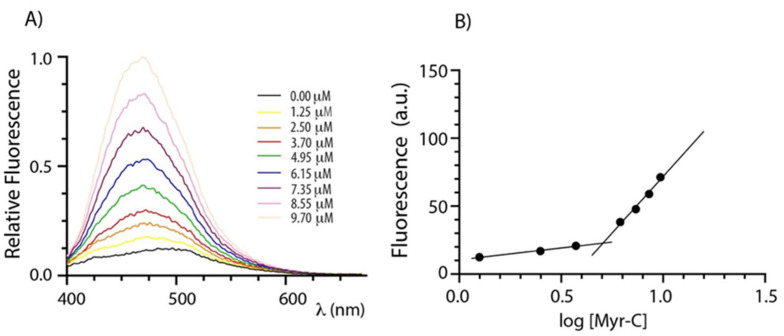
Determination of CMC. (**A**) ANS fluorescence spectra of peptide Myr-C at increasing concentration. (**B**) The plot of ANS fluorescence at 465 nm vs. logarithm of lipopeptide concentration. The intersection of the lines indicates the CMC.

**Figure 2 ijms-23-02164-f002:**
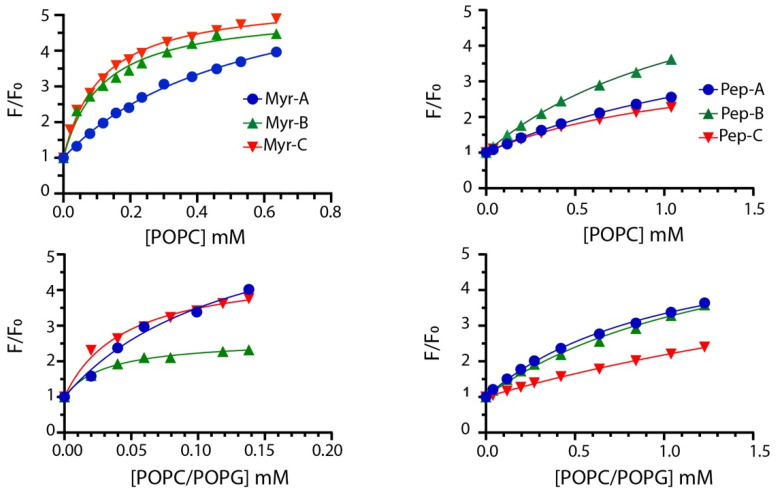
Partition isotherms for myristoylated (**left column**) and non-myristoylated peptides (**right column**) in the presence of zwitterionic and anionic LUVs. The peptide concentration was 1.0 μM. The experiments were performed in quadruplicate, and representative results are shown.

**Figure 3 ijms-23-02164-f003:**
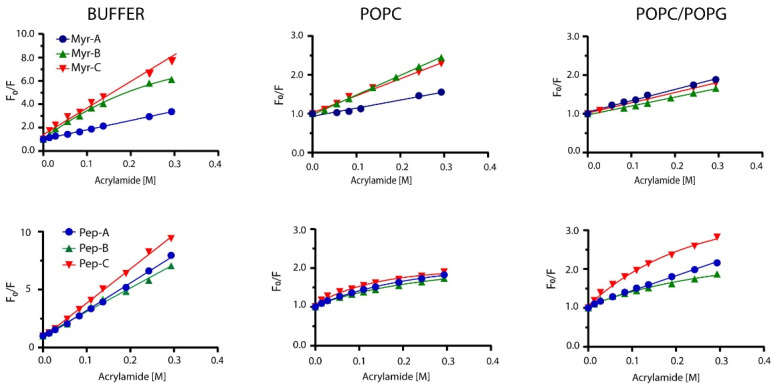
Stern–Volmer plots for the quenching of Trp-1 by acrylamide. Top plots represent myristoylated peptides, whereas bottom plots show non-myristoylated peptides.

**Figure 4 ijms-23-02164-f004:**
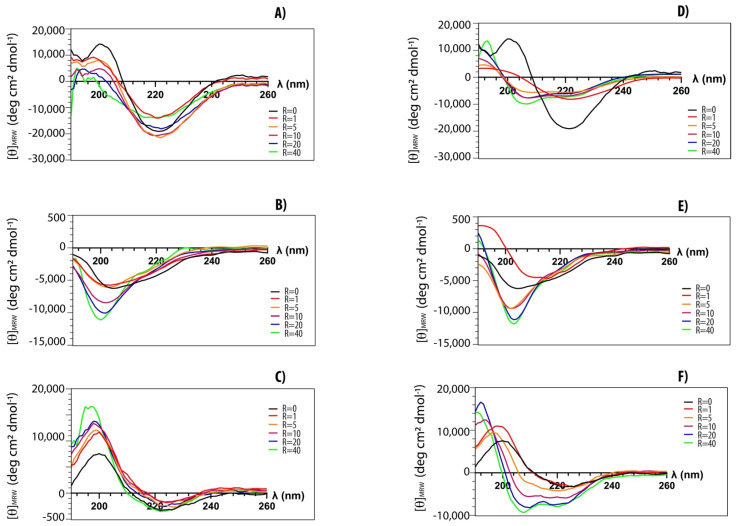
CD-spectra of myristoylated peptides in phosphate buffer (black) and in the presence of increasing amounts of lipid vesicles: (**A**) Myr-A with POPC vesicles, (**D**) Myr-A with POPC/POPG vesicles, (**B**) Myr-B with POPC vesicles, (**E**) Myr-B with POPC/POPG vesicles, (**C**) Myr-C with POPC vesicles, (**F**) Myr-C with POPC/POPG vesicles. The peptide-to-lipid ratio R ranges from 0 to 40.

**Figure 5 ijms-23-02164-f005:**
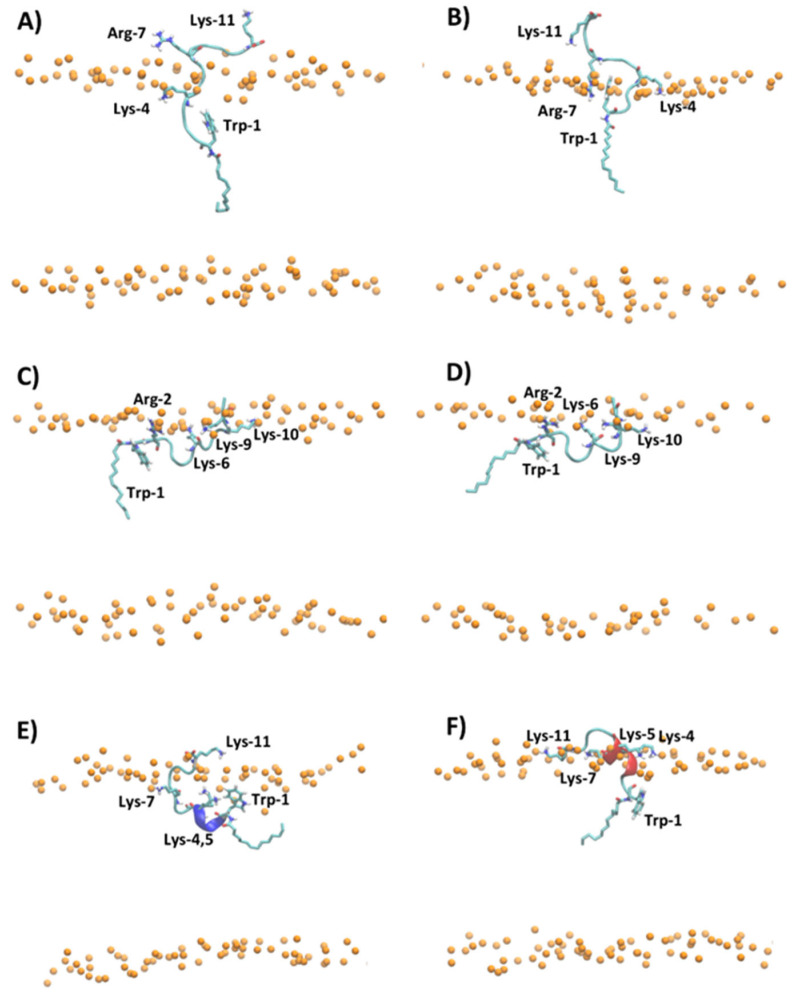
Snapshot of MD simulation at 1 μs of (**A**) Myr-A in POPC, (**B**) Myr-A in POPC/POPG, (**C**) Myr-B in POPC, (**D**) Myr-B in POPC/POPG, (**E**) Myr-C in POPC, and (**F**) Myr-C in POPC/POPG. The phosphorous atoms of lipids are represented as orange dots. The residues of lysine, arginine, tryptophan, and myristoyl chain are represented as sticks, whereas the hydrophobic amino acids are represented as orange sticks. The water and counterions are omitted for clarity.

**Figure 6 ijms-23-02164-f006:**
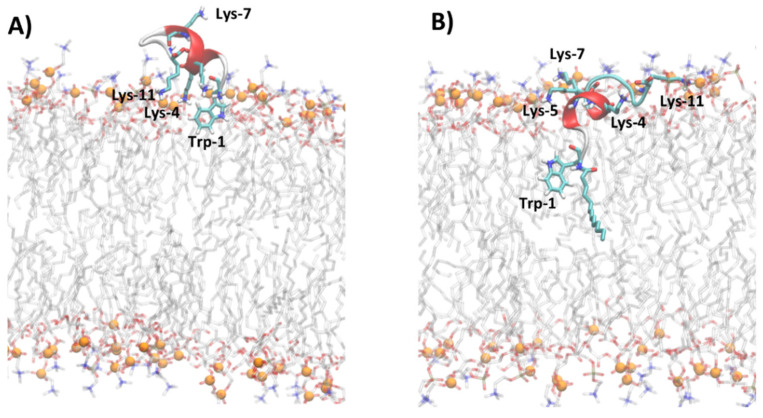
Snapshot of MD simulations of Pep-C (**A**) and Myr-C (**B**) in POPC/POPG. The secondary structure of peptides is represented in cartoon and α-helix is colored in red, while lysine, tryptophan, and myristoyl chains are represented as sticks. The lipid molecules are represented as thin sticks and colored according to their atom types (red for oxygen, blue for nitrogen, orange for phosphorous atoms, grey for carbon atoms of POPC and POPG). Phosphorous atoms of lipid molecules are represented as orange spheres. The water and counterions are omitted for clarity.

**Figure 7 ijms-23-02164-f007:**
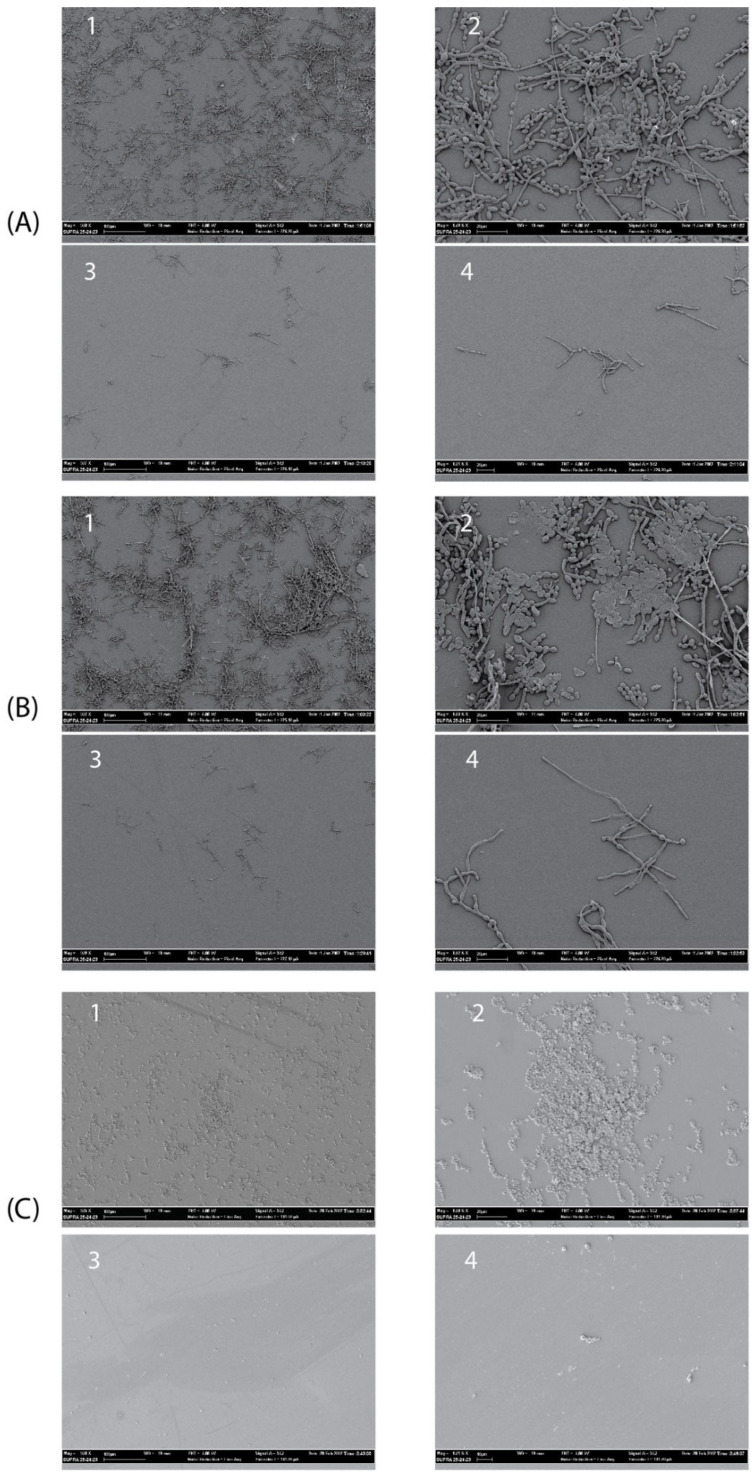
SEM images showing the details of treated and untreated Candida biofilms. (**A1**,**A2**) Untreated *C. albicans* biofilm (magnification 500× and 1800×, respectively); (**A3**,**A4**) treated *C. albicans* biofilm (magnification 500× and 1800×, respectively) with peptide Myr-B 128 µg/mL. (**B1**,**B2**) untreated *C. tropicalis* biofilm; (**B3**,**B4**) treated *C. tropicalis* biofilm with peptide Myr-B128 µg/mL. (**C1**,**C2**) untreated *C. auris* biofilm; (**C3**,**C4**) treated *C. auris* biofilm with peptide Myr-B 128 µg/mL (85 µM).

**Figure 8 ijms-23-02164-f008:**
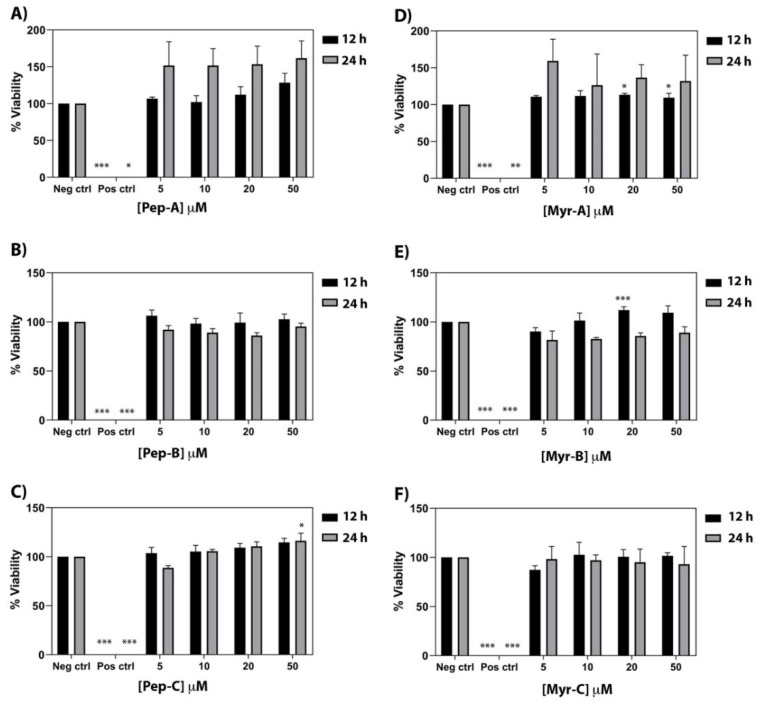
Cytotoxic activity of the six peptides against a primary human fibroblast cell line (FB789). (**A**–**C**) Non myristoylated peptides; (**D**–**F**) Myristoylated peptides. Four different concentrations and two time points have been tested. The values represent the mean + SD (*n* = 3). The asterisks indicate the significance level with respect to negative control (0% of toxicity): *** = *p* < 0.001; ** = *p* < 0.005; * = *p* < 0.05.

**Figure 9 ijms-23-02164-f009:**
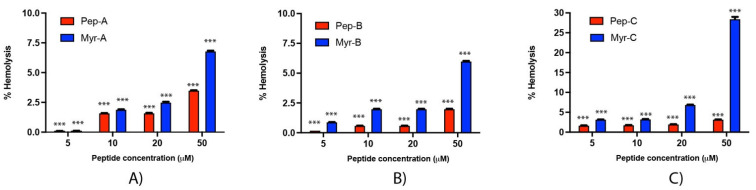
Hemolytic activity of the six peptides against rabbit erythrocytes (**A**) Pep-A and Myr-A; (**B**) Pep-B and Myr-B; (**C**) Pep-C and Myr-C. Four different concentrations have been tested. The values represent the mean + SD (*n* = 3). The asterisks indicate the significance level with respect to positive control (100% of hemolysis): ***= *p* < 0.001.

**Table 1 ijms-23-02164-t001:** Physico-chemical properties of peptides and lipopeptides from the Chionodracine mutant KHS-Cnd.

Peptide Name	Sequence	Net Charge at Neutral pH	Molecular Weight (Da)
KHS-Cnd	WFGKLYRGITKVVKKVKGLLKG	7	2519.16
Pep-A	WFGKLYRGITK	3	1368.66
Pep-B	WRGITKVVKKV	4	1313.66
Pep-C	WVVKKVKGLLK	4	1297.71
Myr-A	Myr-WFGKLYRGITK	2	1579.04
Myr-B	Myr-WRGITKVVKKV	3	1524.04
Myr-C	Myr-WVVKKVKGLLK	3	1508.09

**Table 2 ijms-23-02164-t002:** Partition parameters in the presence of LUVs of different compositions.

Peptide	Lipid Composition	*K_X_*	ΔG (kJ/mol)	Selectivity Ratio
Myr-A	POPC	(1.03 ± 0.04) × 10^5^	−28.58	4.17
	POPC/POPG	(4.3 ± 0.8) × 10^5^	−32.15	
Myr-B	POPC	(4.7 ± 0.6) × 10^5^	−32.37	3.62
	POPC/POPG	(1.7 ± 0.2) × 10^6^	−35.56	
Myr-C	POPC	(4.9 ± 0.3) × 10^5^	−32.44	2.45
	POPC/POPG	(1.2 ± 0.2) × 10^6^	−34.70	
Pep-A	POPC	(3.0 ± 0.1) × 10^4^	−25.55	1.73
	POPC/POPG	(5.2 ± 0.4) × 10^4^	−26.92	
Pep-B	POPC	(4.2 ± 0.2) × 10^4^	−26.39	0.91
	POPC/POPG	(3.8 ± 0.6) × 10^4^	−26.14	
Pep-C	POPC	(4.4 ± 0.2) × 10^4^	−26.50	0.30
	POPC/POPG	(1.3 ± 0.2) × 10^4^	−23.48	

**Table 3 ijms-23-02164-t003:** Stern–Volmer constant (*K_sv_* and K*_a_*), fraction accessible to the quencher (*f_a_*), and Net Accessibility Factor (NAF) for myristoylated peptides in the presence/absence of vesicles of different compositions. NAF and *fa* are defined in the text.

Peptide	LUV Composition	*K_SV_* (M^−1^)	*f_a_*	K*_a_* (M^−1^)	NAF
Myr-A	Buffer	8.0 ± 0.1	1		
	POPC	2.1 ± 0.2	1		0.26
	POPC/POPG	2.9 ± 0.1	1		0.36
Myr-B	Buffer		0.92	34.7 ± 0.6	
	POPC	5.1 ± 0.1	1		0.15
	POPC/POPG	2.3 ± 0.1	1		0.07
Myr-C	Buffer	20.5 ± 1	1		
	POPC	4.4 ± 0.1	1		0.21
	POPC/POPG	2.7 ± 0.1	1		0.13
Pep-A	Buffer	23.7 ± 0.4	1		
	POPC		0.50	14.3 ± 0.1	0.60
	POPC/POPG		0.55	14.5 ± 0.2	0.61
Pep-B	Buffer	21 ± 0.1	1		
	POPC		0.36	21 ± 0.2	1.00
	POPC/POPG		0.44	17.0 ± 0.1	0.81
Pep-C	Buffer	38.8 ± 0.6	1		
	POPC		0.44	35 ± 0.1	0.90
	POPC/POPG		0.72	21 ± 0.1	0.54

**Table 4 ijms-23-02164-t004:** MIC_90_ values of peptides and lipopetides in μg/mL against different *Candida* spp.

			MIC μg/mL (Range)			
*Candida* spp.	Pep-A	Myr-A	Pep-B	Myr-B	Pep-C	Myr-C
*C. albicans*	>256	16 (8–16)	>256	16	>256	32
*C. glabrata*	>256	16	>256	32 (16–32)	>256	32
*C. parapsilosis*	>256	16	>256	32	>256	32
*C. tropicalis*	>256	16	>256	8	>256	16
*C. auris*	>256	>256	>256	16 (16–32)	>256	64 (16–64)

**Table 5 ijms-23-02164-t005:** In vivo toxicity data.

			DAY 1		DAY 2		DAY 3	
	Strain	Peptide	Dead	Alive	Dead	Alive	Dead	Alive
Peptide toxicity	-	Pep-B	3	7	5	5	9	1
	-	Myr-B	1	9	1	9	1	9
Peptide efficacy	*Candida auris*	Pep-B	7	3	10	0	10	0
		Myr-B	2	8	7	3	7	3
Control *	*Candida auris*	-	6	4	7	3	7	3

Note. * The control group larvae were infected with the pathogen but not treated with peptides.

## Data Availability

The data are available on request from the corresponding author.

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
