# Peer review of "Design and Characterization of Myristoylated and Non-Myristoylated Peptides Effective against Candida spp. Clinical Isolates"

_ijms, 2022, doi:10.3390/ijms23042164_

Round 1

Reviewer 1 Report

In this work the authors investigate the antifungal activity of three designed myristoylated and non-myristoylated peptides against Candida spp in vitro and in vivo.

The manuscript is well written and the data presented supports the result that myristoylated peptide is a potential candidate to develop antifungal agents against human fungal pathogens. The results are clearly and informative.

Minor comments to the authors are as follows:

Please add properties of parental peptide, i.e., KHS-Cnd in Table 1. What’s the meaning of value in parentheses in Table 1?

No CMC data of peptide Myr- A and B were available, please provide as supplementary.

In Fig 2, why the tested concentration of LUVs is different for myristoylated and non-myristoylated peptides?

In Fig 4, please indicate the peptide to lipid ratio of each color line.

Table 4, What’s the meaning of no. after each fungi species, such as C. albicans (10)? Please delete them.

How many independent replicates were carried out for in vivo toxicity and efficacy of peptide? 400 µM of Peptide-B is toxicity to insects, why is it still used in next efficacy assay? The concentration tested in Myr-B efficacy is 20-fold of MIC for C. auris. However, the efficacy of peptide is similar with negative control. How to explain this?

Please do statistical significance analysis in Fig 8 and 9.

Author Response

We are pleased with the positive feedback of the reviewer. Below are responses to all the comments and concerns.

REVIEWER #1

Please add properties of parental peptide, i.e., KHS-Cnd, in Table 1. What’s the meaning of value in parentheses in Table 1?

AU: The properties of the parent peptide are now reported in Table 1. We dropped the values in parenthesis corresponding to the measured mass spectrum peak.

No CMC data of peptide Myr- A and B were available, please provide as supplementary.

AU: As suggested, we added Figure S-1 and Figure S-2 in supplementary materials along with the CMC data for both  Myr-A and Myr-B.

In Fig 2, why the tested concentration of LUVs is different for myristoylated and non-myristoylated peptides?

AU: The differences in LUV’s concentration are due to the different partition constant (Kx) values for myristoylated and non-myristoylated peptides. The Kx of myristoylated peptides are about 30-90 folds those of the non-myristoylated peptides.

In Fig 4, please indicate the peptide to lipid ratio of each color line.

AU: As requested, we modified the figure reporting the L/P ratio.

Table 4, What’s the meaning of no. after each fungi species, such as C. albicans (10)? Please delete them.

AU: As suggested, we dropped the (10)

How many independent replicates were carried out for in vivo toxicity and efficacy of peptide? 400 µM of Peptide-B is toxicity to insects, why is it still used in next efficacy assay? The concentration tested in Myr-B efficacy is 20-fold of MIC for C. auris. However, the efficacy of peptide is similar with negative control. How to explain this?

AU: The experiments were repeated at three different times, and each set consisting of 10 larvae. As specified in the text, larvae were treated with a single dose of the peptide, which was effective in counteracting the C. auris infection in the first 24h (treated group showed 20% mortality vs control group which showed 60% mortality). The table shows that this difference is canceled out on the second and third days. Further investigations are necessary to explain the trend; we think that metabolism might probably neutralize the peptide by proteases or other inactivating systems,  allowing the fungal cells to replicate and kill the larvae.

Please do statistical significance analysis in Fig 8 and 9.

AU: We now report the statistics in Figure 8 and Figure 9.

Reviewer 2 Report

This ms. of Bugli et al. contains interesting information on physicochemical properties and antifungal in vitro and in vivo activity of three 11 amino acids-long oligopeptides, myristoylated at the N-terminus. The authors characterised properties of these compounds in solution, modelled their behaviour in membrane and demonstrated their good antifungal activity in vitro and in Galleria mellonella in vivo model of C. auris infection and little if any mammalian toxicity. This is a good, well written work and should be published in IJMS after some minor revision.

Particular comments

1. The title seems to be too general, since it suggests that all or majority of Candida spp. strains susceptible to myristoylated peptides were of the MDR phenotype. However, from the description presented in 4.6.1 it is clear that such phenotype could be attributed probably only to C. auris strains, which are intrinsically resistant to Fluconazole. So that, either the missing information on possible MDR phenotype of the Candida spp. strains tested in this work will be included or the title will be modified. 

2. The obvious error in Table 4 should be corrected. Concentration values given in brackets are μM, not mM. Nonetheless, these values in the 5 - 42 μM range, are above the CMC values determined for myristoylated peptides (4.7 - 5.1μM). One could expect some problems with compounds' solubility at concentrations exceeding CMCs, This issue should be commented on by the authors.

3. The MIC values were determined against 10 clinical strains of 5 different Candida spp., while in Table 4 the single values are presented. The authors should make clear, whether  in each case the detemined MICs were exactly the same for all 10 strains, or the presented numbers are the means of 10 different values. If the latter is true, it would be more informative to present the MIC range instead of a single value.

4. The text is generally well written and only some polishing would be welcome. Please note that a peptide cannot be "absorbed" on the surface (line 316). "Adsorbed" is a proper expression in this context.

Author Response

We are pleased with the positive feedback of the reviewer. Below are responses to all the comments and concerns.

The title seems to be too general, since it suggests that all or majority of Candida spp. strains susceptible to myristoylated peptides were of the MDR phenotype. However, from the description presented in 4.6.1 it is clear that such phenotype could be attributed probably only to C. auris strains, which are intrinsically resistant to Fluconazole. So that, either the missing information on possible MDR phenotype of the Candida spp. strains tested in this work will be included or the title will be modified

AU: As correctly suggested we removed from the title Multidrug-resistant

The obvious error in Table 4 should be corrected. Concentration values given in brackets are μM, not mM. Nonetheless, these values in the 5 – 42 μM range, are above the CMC values determined for myristoylated peptides (4.7 - 5.1μM). One could expect some problems with compounds' solubility at concentrations exceeding CMCs, This issue should be commented on by the authors.

AU: We corrected table 4 according to the referee’s suggestion.  All myristoylated peptides have a solubility of 1mg/mL at 25°C.

On line 665 we added the following sentence:

“In these range of concentration lipopeptides were completely soluble.”

 Moreover, we also added some comments on lines 397-404.

“MIC values are higher than CMC, so MIC values are determined in the presence of lipopetide micellar aggregates and not the free monomer. Under these conditions, lipopetides are still soluble even if in the form of a micellar aggregates. Thus, the activity of lipopetides might also depend on the so-called micellar mechanism consisting in the solubilization of part of the host membrane into mixed micelles. However, further studies are needed to verify this hypothesis”

The MIC values were determined against 10 clinical strains of 5 different Candida spp., while in Table 4 the single values are presented. The authors should make clear, whether  in each case the detemined MICs were exactly the same for all 10 strains, or the presented numbers are the means of 10 different values. If the latter is true, it would be more informative to present the MIC range instead of a single value.

AU: The MIC range has been added in Table.4 . In addition, the table with the individual MICs of each isolate of Candida species have been added in the Supplementary Information (Table S1).

The text is generally well written and only some polishing would be welcome. Please note that a peptide cannot be "absorbed" on the surface (line 316). "Adsorbed" is a proper expression in this context. 

AU: We agree with the reviewer. We changed ‘absorbed’ with ‘adsorbed’ and corrected other misspellings.